# CAN GENERATIVE MULTIMODAL MODELS COUNT TO TEN?

**Sunayana Rane**
Department of Computer Science
Princeton University
srane@princeton.edu

**Alexander Ku**
Google DeepMind
alexku@google.com

**Jason Baldridge**
Google DeepMind
jasonbaldridge@google.com

**Ian Tenney**
Google Research
iftenney@google.com

**Thomas L. Griffiths\***
Departments of Psychology and Computer Science
Princeton University
tomg@princeton.edu

**Been Kim\***
Google DeepMind
beenkim@google.com

## ABSTRACT

We adapt a developmental psychology paradigm to characterize the counting ability of the foundation model Parti. We show that three model scales of the Parti model (350m, 3B, and 20B parameters respectively) each have *some* counting ability, with a significant jump in performance between the 350m and 3B model scales. We also demonstrate that it is possible to interfere with these models' counting ability simply by incorporating unusual descriptive adjectives for the objects being counted into the text prompt. We analyze our results in the context of the knower-level theory of child number learning. Our results show that we can gain experimental intuition for how to probe model behavior by drawing from a rich literature of behavioral experiments on humans, and, perhaps most importantly, by adapting human developmental benchmarking paradigms to AI models, we can characterize and understand their behavior with respect to our own.

## 1 BACKGROUND AND MOTIVATION

With text-to-image multimodal models gaining widespread use, it is more important than ever to characterize and methodically study their behaviors. Recent research has focused on studying whether these models demonstrate compositionality, appropriately producing the right combination of abstract concepts (Thrush et al., 2022). Here we focus on an even simpler criterion for abstraction: understanding number. For example, we might wonder whether a model can reliably count to ten, and whether its internal understanding of number concepts matches what we would expect of a human. Many of the questions we are now asking of a model's "understanding" of concepts are the same questions we've asked previously about human children (Frank, 2023a;b). Developmental cognitive psychologists have devised tests and measures that probe many aspects of a child's understanding of number. For example, researchers discovered that children often produce (speak) number words in order ("one, two, three") before they understand how to use them or what they mean (before they can produce 3 objects, or even correctly count the 3 objects placed in front of them) Sarnecka & Carey (2008); Frye et al. (1989); Fuson (2012). Here we adapt these procedures to study multimodal models.

Standard evaluations of multimodal models focusing on a broad set of capabilities also often include some measure of counting ability among a larger set of metrics (Cho et al., 2023a;b; Hu et al., 2023; Lee et al., 2023). In contrast to these broader metrics designed to provide a standard measure of a wide range of model abilities, here we provide a deeper behavior-based analysis which systematically and specifically varies counting along controlled lines.

## 2 TASK AND DATA

The task often used as the gold-standard for measuring a child's understanding of number concepts is known as the Give-N task (Frye et al., 1989; Fuson, 2012; Schaeffer et al., 1974; Wynn, 1990; 1992; Marchand et al., 2022). The idea is simple: prompted with an instruction like "give five lemons," the child must physically count out and give five toy lemons to a puppet. Instead of other tasks that ask children to count sets of objects they are given (as in the How-Many task), the Give-N task is understood to provide a very rigorous standard for number understanding; many children who can verbally count up to 5 cannot successfully produce sets of 5 items, so their performance on the Give-N task indicates that they *thoroughly* understand a number (Wynn, 1990; 1992).

Generative vision and language models can now be probed using something similar to the Give-N task, prompted with text like "five lemons" and asked to generate an image from scratch. This is in contrast to classification or captioning models, in which you can only be asked to count the number of objects in an input image, corresponding to the more lax "How-Many" task used in developmental psychology (Wynn, 1992; Connor et al., 2024). We gave the Parti model text prompts based on the Give-N task and engaged a pool of human raters through a crowdsourcing contractor to count objects in the generated images.

### 2.1 STANDARD PROMPTS

To approximate the Give-N task in a way that makes sense with the modalities of our models, we start with child word learning data. The "object" words used to create the prompts of our version of the Give-N task were the 40 most easily learned food and animal words (20 of each) of all the words in the WordBank database of child vocabulary development (Frank et al., 2017). These are words most children learn prior to 36 months, and are close analogues to the food and animal toys in the original Give-N experiments. We give each model prompts of these 40 objects, with counts from 1-15.

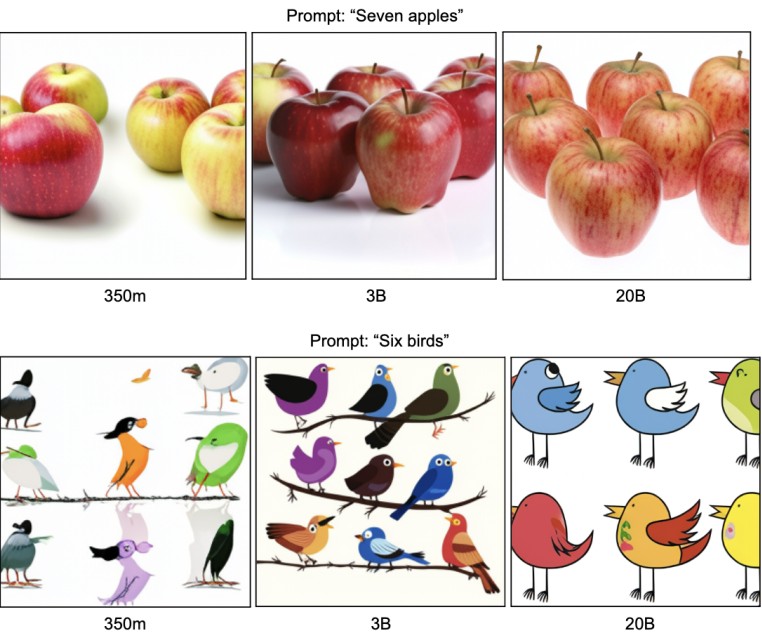

Figure 1: Images generated by each of the three scales of Parti models for the input text prompts "seven apples" and "six birds".

## 2.2 COMMON AND UNCOMMON ADJECTIVE PROMPTS

In addition to the standard prompts, we also constructed a smaller set of prompts to probe how dependent counting performance is on the familiarity of the objects in the prompt – in other words, can we interfere with counting ability by modulating other things about the prompt? We refer to these as the common and uncommon prompts, fully listed in Table 1. We use a subset of the objects used in the standard prompts: apples, oranges, bananas, cats, and dogs. For each of these object types, we modulate an adjective: we use one that is common for the object, such as "fluffy" for dogs, and one that is uncommon, such as "spiky" for dogs.

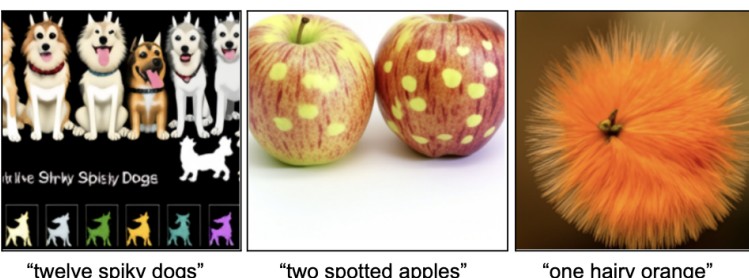

"twelve spiky dogs"   "two spotted apples"   "one hairy orange"

Figure 2: Examples of images generated for the common and uncommon adjective prompts. These outputs were generated by the Parti 3B model.

Table 1: Common and uncommon prompts

| Common prompts | Uncommon prompts |
| --- | --- |
| "six red apples" | "six spotted apples" |
| "two black cats" | "two green cats" |
| "four yellow bananas" | "four blue bananas" |
| "three fluffy dogs" | "three spiky dogs" |
| "eight shiny oranges" | "eight hairy oranges" |

## 3 MODELS

One way to characterize a developmental trajectory in models is to explore several different scales of one model type. The Pathways Autoregressive Text-to-Image (Parti) model (Yu et al., 2022) provides just such an opportunity, because it presents a common model architecture at multiple scales. For our experiments we use three different scales of the Parti model: 350M, 3B, and 20B parameters. The model architecture follows a Transformer-based (Vaswani et al., 2017) encoder-decoder framework, with the decoder receiving the major share of the increase in each increasingly large model size. The Parti models take text input such as "five lemons" and output generated images.

## 4 RESULTS

For each of the prompt categories (standard, common, and uncommon prompts) we provide a detailed breakdown of the correlation between the number of objects in the model-generated images (as recorded by human evaluators) and the true count from the text prompt the model was given (Table 2). In addition, figure 3 gives a detailed breakdown of each model's count-by-count performance on the standard prompts.

Table 2: Correlations between model-generated image count and true input count from text prompt

|  | Model | Pearson | p-value | Spearman | p-value |
|---|---|---|---|---|---|
| Standard prompts | 350m | 0.4071 | 2.5601e-25 | 0.4735 | 8.4204e-35 |
|  | 3B | 0.6647 | 1.6888e-77 | 0.7159 | 4.6031e-95 |
|  | 20B | 0.6781 | 6.9107e-82 | 0.7099 | 5.7467e-93 |
| Common prompts | 350m | 0.6334 | 1.5229e-12 | 0.6873 | 2.8497e-15 |
|  | 3B | 0.7768 | 2.1413e-21 | 0.7810 | 9.5063e-22 |
|  | 20B | 0.6458 | 4.0101e-13 | 0.6648 | 4.6304e-14 |
| Uncommon prompts | 350m | 0.2882 | 3.6466e-03 | 0.1438 | 1.5356e-01 |
|  | 3B | 0.4862 | 2.9217e-07 | 0.5062 | 7.7494e-08 |
|  | 20B | 0.4734 | 6.5363e-07 | 0.4508 | 2.5144e-06 |

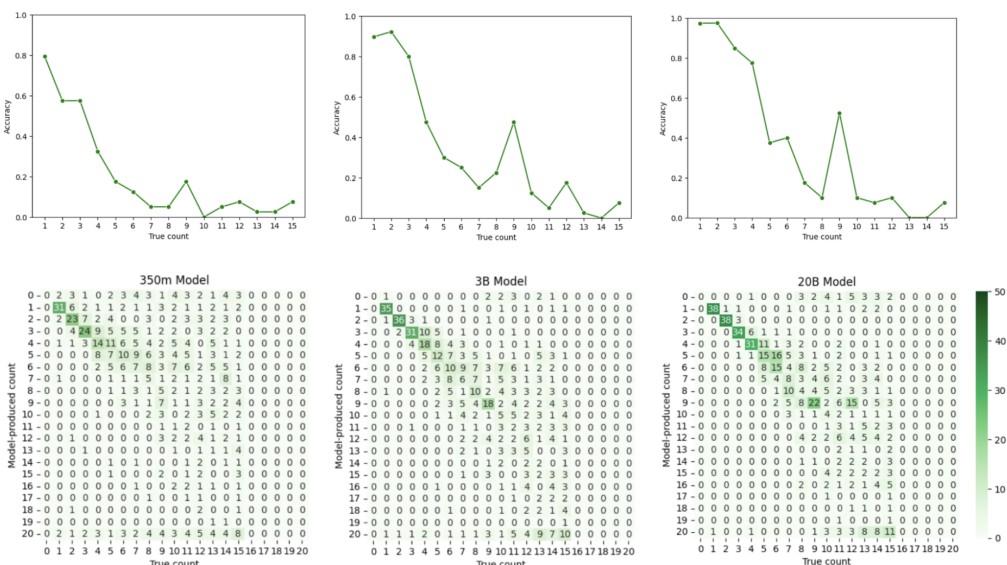

Figure 3: Standard prompt results in greater detail: counts of objects in images generated by each of the three scales of Parti model, compared to the true count as specified in the text prompt.

## 5 DISCUSSION

We show results of the Give-N task from developmental psychology, as adapted to evaluate the large multimodal model Parti at three different model scales. Our results show that all three scales of the Parti model have *some* counting ability, with a steep increase in performance between the 350m model and the 3B model. Both the 3B model and 20B model have similar performance in most categories, indicating that counting skills may be "unlocked" at the 3B model scale.

In analyzing the results, we also draw from the knower-levels framework often used to understand Give-N task results in child psychology studies (Wynn, 1992; Sarnecka & Carey, 2008). At the "one-knower" level, which most children reach by 2.5-3 years, they understand only the concept of 1. A few months later, a child becomes a "two-knower," when they reliably give 1 and 2, but not 3, 4, 5. Then slowly comes the "three-knower" and, some studies report, the "four-knower" level before the child learns the subsequent numbers not slowly, as before, but all at once and through induction (they have learned that adding one to a prior number results in the next number).

Interestingly, our results suggest that this inductive step is missing from all three scales of the models' behavior. The 20B model seems to be inching along in this direction, getting fairly reliable results up to 4 and improved results on 5 and 6 compared to both the 350m and 3B models. However,

this behavior obviously has not scaled past 5-6, and from 7 onward it is quite difficult for the models, in contrast to children who learn 5 onward quickly and inductively. Our approach illuminates this gap, and shows behavioral similarities between these models and children of approximately 3-4 years of age.

Furthermore, the results for the common and uncommon adjective prompts demonstrate a significant gap in performance across all models, indicating that unusual adjective and noun combinations do indeed interfere with the model's counting ability. This highlights an area for further training, investigation, and improvement in these models. There are also areas of strength: notably, the 3B model demonstrates particularly strong performance for common adjective prompts, surpassing even the 20B model and painting a more nuanced picture of how model scale relates to performance.

We hope this approach empowers model designers to address developmental gaps in knowledge and performance, and that the practice of using developmental psychology paradigms to probe model behavior continues to help us develop more reliable, responsible foundation models.

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
