# OpenReview forum: "Can Generative Multimodal Models Count to Ten?"
_ICLR.cc/2024/Workshop/Re-Align — ICLR 2024 Workshop Re-Align Poster_

### Official Review · Reviewer_4WaV · 2024-02-18
**Nice short paper using developmental approach reveals fundamental gap between models and humans**

**Rating:** 2
**Fit:** 3
**Confidence:** 2

**Workshop Review:**

Summary:
This paper investigates the alignment between text-to-image multimodal AI models and human children, focusing on the task of understanding numbers. Despite its seemingly straightforward nature, the authors demonstrate that large-scale multimodal models, such as Parti, struggle to generate images that accurately represent the quantity of objects specified in the text prompt. In a broader context, the paper suggests the importance of addressing developmental gaps in both knowledge and performance when developing foundational models. It advocates for the use of developmental psychology paradigms as a reliable approach to probe model behaviors and facilitate the development of more aligned models.

Overall, I find this paper to be of interest to the alignment research community. It is well-written, easy to follow, and the experiments are logically designed. I particularly like the authors' approach of prompting models to generate images as a means of testing their understanding of numbers, which offers advantages over methods such as counting objects in input images. Additionally, the experiment involving common and uncommon adjective prompts yields intriguing results, and I would welcome further insights from the authors into the underlying reasons for these findings.

I recommend the acceptance of this short paper at ReAlign.

**Reason For Not Giving Higher Score:**

N/A

**Reason For Not Giving Lower Score:**

N/A

**Reviewer Domain:**

cognitive science

---

### Official Review · Reviewer_fEKp · 2024-02-22
**Interesting toy study**

**Rating:** 2
**Fit:** 2
**Confidence:** 2

**Workshop Review:**

The paper investigates the counting ability of a generative multimodal large language model at different scales.


**Clarity**

I found the paper to be engaging and easy to read. However, I'm left wondering about the take-away: generative large language models do not learn counting like children? Is your point that as these models scale, they do not fit into the knower-level framework, and are thus unlike children in how they learn numbers? I think this could be made a bit more clear.

**Correctness**

Overall, I think the approach makes sense and the methods seem correct. However, there remain a few points I would like to have more information on:
- In 4 Results, I would like to know more details about the human evaluators (how many for instance).
- In Table 2, how do you explain that the 3B model correlates higher with the true input count compared to the 20B model? This seems counterintuitive to me.
- In Figure 3, how do you explain the spike at true count = 9?
- In Figure 3 second row, how come not all columns add up to 40 (for instance true count = 1)?
- In 5 Discussion: ''indicating that counting skills may be “unlocked” at the 3B model scale.'' seems like a stretch, since all models still perform poorly for most numbers above 4. Also [1] show that multimodal large language models with 8B parameters still struggle with counting and that even GPT-4V struggles for some data sets where there are more than 10 objects in a scene (so likely it isn't unlocked at the 3B model scale).

**Novelty**

As far as I can tell this is a novel investigation. There are papers that investigate multimodal large language models' counting abilities (such as [1]) but not for generative multimodal large language models.

**Interest to community**

This paper is of interest to the research community in general but since it does not investigate representations I am not sure if it is of particular interest for visitors of the Re-Align workshop.

[1] Schulze Buschoff, Luca M., et al. "Have we built machines that think like people?." arXiv e-prints (2023): arXiv-2311.

**Reason For Not Giving Higher Score:**

I outlined most reasons in the sections above. Additionally, since the paper doesn't actually focus on representations themselves but rather on the output of the generative model, I am unsure about the fit to the workshop.

**Reason For Not Giving Lower Score:**

Overall, while the scope of the paper is rather small, the paper is well-written and briefly outlines an interesting investigation. In any case, I feel like it is of some interest to the community as a whole and I would therefore recommend acceptance.

**Reviewer Domain:**

machine learning

---

### Official Review · Reviewer_G7bT · 2024-02-29
**Can Generative Multimodal Models Count to Ten?**

**Rating:** 1
**Fit:** 3
**Confidence:** 2

**Workshop Review:**

This paper frames the evaluation of counting ability in multimodal language-vision models from the lens of developmental psychology, where two different tasks are used to evaluate counting ability in infants: the "how many" task, analogous to image captioning (or image-to-text), and the "give N" task, analogous to image generation (text-to-image). There are interesting learning dynamics when human children ages 2.5-5 learn to count, and this motivates the authors to compare 3 different text-to-image models with varying size. Prior work considers counting ability in language-vision models as part of a broader evaluation suite, whereas this work contributes a more in-depth analysis of counting ability by systematically testing ability for counting numbers 1-15 with common entities in images and language corpora such as birds and apples. Their main conclusions from their results are:
1. "all three scales of the Parti model have some counting ability, with a steep increase in performance between the 350m model and the 3B model ... counting skills may be “unlocked” at the 3B model scale"
2. "this inductive step is missing from all three scales of the models’ behavior ... Our approach illuminates this gap, and shows behavioral similarities between these models and children of approximately 3-4 years of age."

I believe this work is well-motivated, as counting objects is often discussed as a domain for evaluating text-image models. I appreciated the analogies with developmental psychology and the "give N" vs. "how many" framing seems novel. The experimental stimuli are reasonable, and the common vs. uncommon adjective comparison is useful. The authors' empirical results in Figure 3 are interesting. I'm not familiar enough with the specific topic of evaluating counting in multimodal models to say for certain if this is the first work to run a correlation analysis between model-generated numbers of objects and what's specified in the prompt, but if so, this is a valuable contribution.

However, I think this paper has a few key weaknesses:
1. It's unclear to me whether some of the authors conclusions follow from their results, based on my knowledge of the relevant literature. For the behavioral similarities with children of 3-4 years of age, how do these results suggest that these models are acting like 5/6-knowers? In the heatmaps of Figure 3, shouldn't we see uniform probabilities across all model-generated numbers over some N-knower threshold?
2. There is no comparison between "give N" and "how many" tasks, i.e. comparing evaluating text-to-image counting vs. image-to-text counting. Since so much of the text in Sections 1 & 2 is about this distinction, and since there is prior work evaluating counting in text-to-image models ("give N" task; e.g. [1]), I'm not sure what to make of this point.
3. Needs at least a few sentences summarizing the most relevant prior work. There's a lot of text on dev psych background, but comparatively little about "What did previous counting evaluations of multimodal models find? What was missing?"
4. There isn't much discussion of the striking biases shown in Figure 3. What's up with the spike in accuracy around 9 for all 3 models? What about the models' bias towards producing "20" for any count above ~10? What do you make of the accuracy for numbers 4-6 being ~30-50% - is there a similar pattern in children where N-knowers have ~50% accuracy on the number N and ~100% for numbers <N?

Smaller suggestions/comments:
- Table 2: add hit/miss accuracy as a column, to match the y-axis in Fig 3 line plots
- Fig 3: It seems like the rightmost 5 columns of the heatmaps could be dropped - these are all 0 and with (1, 15) this would align better with the line plots.
- It seems like even a very capable text-to-image model should have major limitations counting large numbers of objects due to image resolution and limitations of vision as a modality.


[1] Jaemin Cho, Yushi Hu, Roopal Garg, Peter Anderson, Ranjay Krishna, Jason Baldridge, Mohit Bansal, Jordi Pont-Tuset, and Su Wang (2023). Davidsonian scene graph: Improving reliability in fine- grained evaluation for text-image generation.

**Reason For Not Giving Higher Score:**

It's unclear to me whether some of the authors' conclusions follow from their results. The authors distinguish between "give N" and "how many" tasks, but it's unclear what is the point of making this distinction given the scope of this work. Discussion of prior work on evaluating counting ability in multimodal text-image models is lacking.

**Reason For Not Giving Lower Score:**

Novel results on a specific domain, counting objects, commonly discussed in multimodal AI evaluation. Novel framing of problem grounded in developmental psychology.

**Reviewer Domain:**

cognitive science

---

### Decision · Program_Chairs · 2024-03-02

Accept (Poster)